# New soft tissue data of pterosaur tail vane reveals sophisticated, dynamic tensioning usage and expands its evolutionary origins

Natalia Jagielska[1], Thomas G Kaye[2], Michael B Habib[3], Tatsuya Hirasawa[4], Michael Pittman[5]*

[1]School of GeoSciences, University of Edinburgh, Edinburgh, United Kingdom; [2]Foundation for Scientific Advancement, Sierra Vista, United States; [3]Department of Medicine, University of California, Los Angeles, Los Angeles, United States; [4]Department of Earth and Planetary Science, Graduate School of Science, The University of Tokyo, Tokyo, Japan; [5]School of Life Sciences, The Chinese University of Hong Kong, Shatin, Hong Kong SAR, China

## eLife Assessment

The presented soft tissue data of pterosaur tail vanes represent a **valuable** contribution to ongoing research efforts to decipher the flight abilities of pterosaurs in the fields of paleontology, comparative biomechanics, and bioinspired design. The new methods are **compelling** and give new detail on tail morphology, with a potential to resolve how pterosaurs were able to control and maintain tail stiffness to furnish flight control.

*For correspondence:
mpittman@cuhk.edu.hk

Competing interest: The authors declare that no competing interests exist.

**Abstract** Pterosaurs were the first vertebrates to achieve powered flight. Early pterosaurs had long stiff tails with a mobile base that could shift their center of mass, potentially benefiting flight control. These tails ended in a tall, thin soft tissue vane that would compromise aerodynamic control and efficiency if it fluttered excessively during flight. Maintaining stiffness in the vane would have been crucial in early pterosaur flight, but how this was achieved has been unclear, especially since vanes were lost in later pterosaurs and are absent in birds and bats. Here, we use Laser-Stimulated Fluorescence imaging to reveal a cross-linking lattice within the tail vanes of early pterosaurs. The lattice supported a sophisticated dynamic tensioning system used to maintain vane stiffness, allowing the whole tail to augment flight control and the vane to function as a display structure.

## Introduction

Pterosaurs were the first vertebrates to achieve powered flight (*Palmer, 2017*). The first pterosaurs, the non-pterodactyloids, had long, stiff tails with a mobile base (*Frey et al., 2003*), similar to some dinosaurs like *Velociraptor* (*Persons and Currie, 2012*). Many of these tails end in a soft tissue 'vane' (*Marsh, 1882*; *Döderlein, 1929*; *Frey et al., 2003*; *Figure 1*), which may have contributed to passive stability in flight. A primary role in display has also been suggested (*O'Brien et al., 2018*), given ontogenetic shape changes in the vane and the fact that, unlike most aircraft, flying animals do not need vertical control surfaces to be yaw-stable during turns (*Bowers, 2016*). The vanes have been interpreted as steering aids (*Frey et al., 2003*). The length and stiffening of the tails suggest that

**eLife digest** Long before bats and birds, there were the pterosaurs; the first vertebrates to have ever achieved powered flight, soaring through the skies at the dawn of the Age of Dinosaurs. Early species were modest in size, had toothed jaws and sported long, stiff tails that were mobile at the base and ended in tall, thin, soft tissues proposed to have had flight and display functions. However, these blade-like 'vanes' would have hindered the animals' ability to fly if they had fluttered excessively. How early pterosaurs managed to maintain vane stiffness remains unclear, as these structures are absent in later species and in modern flying vertebrates.

Recently, an imaging technique called laser-stimulated fluorescence imaging has allowed researchers to uncover hidden soft tissue features in fossils, using high-power lasers to generate chemical maps of structures on or just below the surface of a specimen. Jagielska et al. relied on this approach to examine the tail vanes of four high-quality pterosaur fossils. This revealed a unique underlying lattice structure composed of thicker vertical elements interlinked with thinner fibres, which would have conferred stiffness during flight. However, this complex organization also suggests a multifunctional design, making it possible for the vanes to have been used as displays for communication or to attract a mate.

These findings deepen our understanding of early pterosaur flight innovations and the evolutionary pressures that shaped the success of the first powered fliers. By highlighting the value of laser imaging for uncovering soft tissue details in fossilised specimens, the work of Jagielska et al. opens avenues for further investigation into similar tissues, such as flight membranes.

they might have been important in early pterosaurs for control based on mass shifting or inertial control, as purported for terrestrial theropods with convergent tails (*Persons and Currie, 2012*). Such dynamic control could greatly improve maneuverability and/or stability. However, vane fluttering would be extremely costly and destabilising unless the vane was tensioned while under aerodynamic load. Tail vanes feature thick, evenly spaced, internal structures roughly perpendicular to the caudal series (*Döderlein, 1929*), that are said to resemble neural spines and haemal arches (*Marsh, 1882*). These structures are presumed to have minimised fluttering and prevented buckling in the same way that spars, ribs, stringers, and longerons do in airplane wings and tail-fins, but others have proposed that they were flexible and cartilaginous (*Marsh, 1882*), especially since their preserved appearance varies. Here, we use laser-simulated fluorescence (LSF) imaging of *Rhamphorhynchus* specimens from the Upper Jurassic Solnhofen Limestones (*Kaye et al., 2015*; *Pittman et al., 2021*) to investigate the vane's structural properties, explore its usage, its evolutionary origins and the context for its disappearance in later pterodactyloids (*Frey et al., 2003*).

## Results

Over 100 Solnhofen pterosaur fossils were examined for well-preserved tail vanes using an ultraviolet torch. Four exceptional specimens were then imaged under Laser-Stimulated Fluorescence (LSF). Three specimens - NHMUK PV OR 37787, NMS G.1994.13.1 and ROM VP 55352 - exhibited tail vanes under white light, but the vane of NHMUK PV OR 37003 was only visible under LSF. LSF confirmed the soft tissue extent of the vanes and revealed hidden anatomical details, especially in NHMUK PV OR 37003 and 37787 and NMS G.1994.13.1 (*Figure 2A–C*), where vane areas fluoresced pink and white, indicating soft tissue preservation (*Pittman et al., 2021*). Tail vanes are sub-symmetrical and diamond-shaped in NHMUK PV OR 37003 and 37787 and NMS G.1994.13.1 with a length of 700 mm, 750 mm, and 720 mm making up 21%, 22% and 21% of the total tail length of 320 mm, 362 mm, and 348 mm, respectively (*Figure 2*). NHMUK PV OR 37003 and 37787 and NMS G.1994.13.1 have tail vanes occupying a similar proportion of the tail (23% and 22%) as BSP 1907 I 37 (74 mm) and YPM VP 1778 (50 mm). At its widest point, about two-thirds along its length, the vane is 41 mm across in NHMUK PV OR 37003 and 37787 but the widest in NMS G.1994.13.1 (55 mm), even wider than in BSP 1907 I 37 (46 mm) and almost twice as wide as YPM VP 1778 (30 mm). The vanes of NHMUK PV OR 37003 and 37787, NMS G.1994.13.1 and YPM VP 1778 form anterior, lateral and posterior angles of 40°, 45° and 90°; 107°, 113° and 105°; 104°, 100° and 90°; 44°, 107° and 80°, respectively. There is

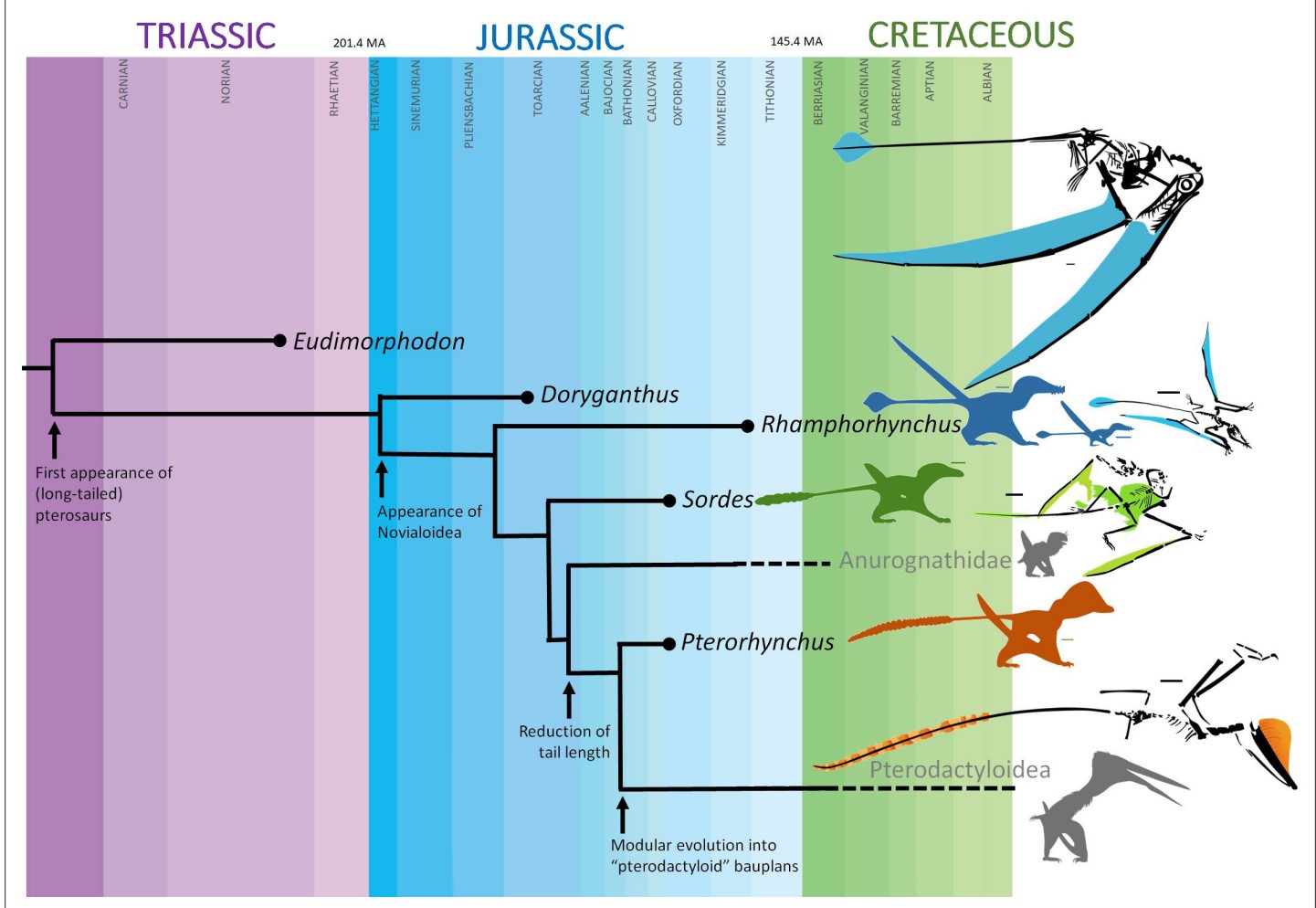

**Figure 1.** Long-tailed early-diverging non-pterodactyloid pterosaurs had diverse tail vanes but these disappeared in later-diverging short-tailed pterodactyloids. Blue, ontogenetic morphs of *Rhamphorhynchus muensteri*: BSPG 1938 I 503 a and ROM VP 55352. Green, *Sordes pilosus* PIN 2585/3. Orange, *Pterorhynchus wellnhoferi* CAGS02-IG-guasa-2/DM608. Scale bars are 3 cm.

no marked change in caudal morphology within and outside the periphery of the tail vane. The vane encompasses at least 15 caudals in NHMUK PV OR 37003 and NMS G.1994.13.1 and at least 17 in NHMUKPOV OR 37787, out of which, 13 appear to bear elongated zygapophyses in 37703 and at least 14 in 37787 (16 caudals within YPM VP 1778 and BSP 1907 I 37). The vane is extrapolated to originate from the anterior end of a caudal in NHMUK PV OR 37003 and 37787 and NMS G.1994.13.1. Under LSF, partial edges of the vane are visible, along with at least 17 relatively straight structures in NHMUK PV OR 37787 (10+in NHMUK PV OR 37003 and 11+ in NMS G.1994.13.1); projecting vertically, near-perpendicularly to the tail skeleton at the widest point of vane (based on position of chevron bones e.g. NHM PV OR 37787), they can project down to acute angles of 57°, 65°, 66°, and 57° in NHMUK PV OR 37003 and 37787, NMS G.1994.13.1 and YPM VP 1778, respectively. In NHMUK PV OR 37003 and OR 37787 and NMS G.1994.13.1, these vertical structures stem from caudal articulation points in the distal portion of the vane but anteriorly there is no obvious pattern. These data do not support their association with the neural spines and haemal arches (*contra **Marsh, 1882***). In NHMUK PV OR 37003 and 37787 and NMS G.1994.13.1, these vertical structures are relatively thick (0.6–1 mm) and appear to be hollow, suggesting they were rod-like, and were arranged in parallel ~3–8 mm apart. In NHMUK PV OR 37003 and 37787 they are rarely preserved dead-straight, but are straighter in NMS G.1994.13.1, especially anteriorly. In YPM VP 1778 and BSP 1907 I 37 the vertical structures show more pronounced undulations giving them a sigmoidal morphology. To our knowledge, in NHMUK PV OR 37003 alone, there is a second layer of thinner and more numerous fibres that run across the thick

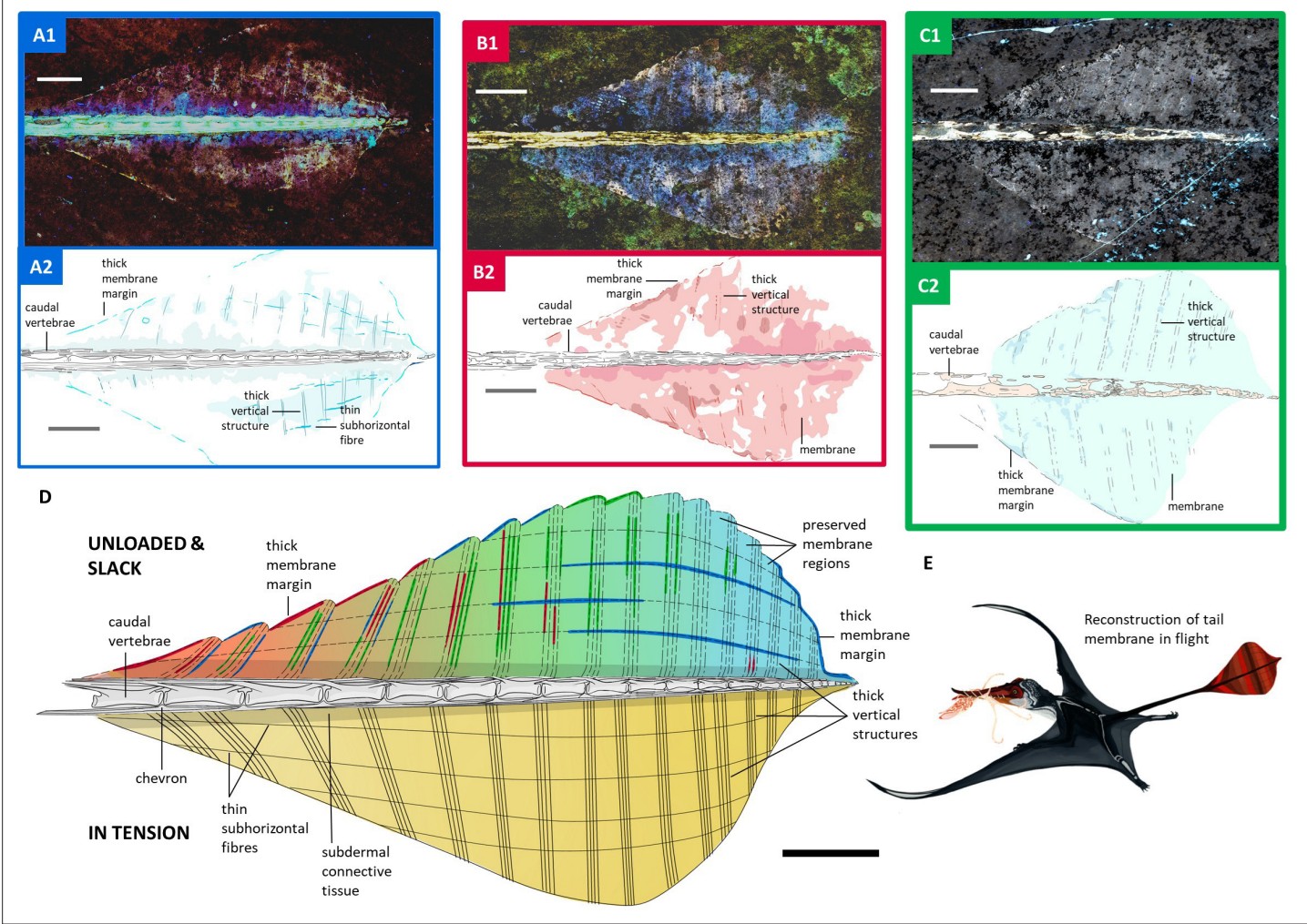

**Figure 2.** Tail vane of *Rhamphorhynchus muensteri*. (**A1**) LSF image of NHMUK PV OR 37787. (**A2**) Line drawing of LSF image of NHMUK PV OR 37787. (**B1**) LSF image of NHMUK PV OR 37003. (**B2**) Line drawing of LSF image of NHMUK PV OR 37003. (**C1**) LSF image of NMS G.1994.13.1. (**C2**) Line drawing of LSF image of NMS G.1994.13.1. (**D**) Interpretative line drawing of *Rhamphorhynchus muensteri* tail vane unloaded and slack as well as in tension. Combines LSF results of NHMUK PV OR 37003 and 37787 as well as NMS G.1994.13.1 (**A1–C2**). (**E**) Life reconstruction of *Rhamphorhynchus muensteri* using its tail vane during flight. All scale bars are 1 cm.

vertical structures, subparallel to the long axis of the tail and become more and more closely spaced as they reach the tail tip. Together, the vertical structures and subhorizontal fibres form a cross-linked lattice. The thick outer margin of the vane is undulated in dorsal view in both NHMUK PV OR 37003 and 37787, with a trough ~0.7 mm deep where the thick vertical structures meet the margin and a convex peak roughly mid-way between each pair of vertical structures. A similarly thick outer margin is also observed anteriorly on the tail vane of NMS G.1994.13.1 and BSP 1907 I 37.

## Discussion

In order to discuss the tail vane's structural properties and their implications, we will first introduce the relevant biomechanical concepts and how they have previously been applied to pterosaur wings. Material stiffness is a measure of a material's resistance to deformation and can be summarised by a variable known as the modulus of elasticity (E) which is the ratio of stress (σ) to strain (ϵ) during deformation. Stress, is the applied force divided by the area over which it is exerted, while strain refers to the change in length of the material: positive strain is stretching (tensile strain) and negative strain is compression (compressive strain). E can differ substantially for different kinds of loading - in soft tissues, resistance to tension is usually much greater than resistance to compression. For instance,

'true' ligaments running between bones typically only load in tension (i.e. when stretched), they essentially just 'fold up' if compressed (*Woo et al., 2007*). Stiffer materials have a higher value of E than flexible materials: it takes more stress to produce the same strain (they deform less for a given stress).

When the soft tissue surface of the pterosaur wing was placed under aerodynamic load, the associated stress was expected to have stretched the wing surface so it curved towards the side of lower air pressure. This resulted in 'auto-cambering' of the wing surface, which became more concave beneath (and more convex above). The degree of auto-cambering was limited by stiffening mechanisms preventing excessive stretching (*Palmer and Dyke, 2012*). Stiffening was provided by a combination of active mechanisms through muscles within the wings as well as passive mechanisms, including keratinisation (stiff keratins have the highest E of any soft tissue) and reinforcement by fibres within the wing (actinofibrils; *Palmer and Dyke, 2012*). Actinofibrils reinforced the wing against stretching by having a comparatively high value of E in tension: as the entire wing surface bowed, both the ventral and dorsal surfaces were loaded in tension, which applied a tensile load to the actinofibrils. The actinofibrils stretched according to their elastic modulus in tension, which limited the stretching of the entire wing surface. This stands in contrast to the cantilever bending of a hard tissue 'beam' such as a long bone, where one side is loaded in compression and the other side is loaded in tension.

The pterosaur tail vane would have experienced asymmetric air flow anytime it was deflected (even slightly) to the right or left during flight. This would produce a moderate amount of lift, oriented laterally, to either the right or left side depending on the motion of the tail. While this modest lift would have been of little use for flight control, it would have had a tendency to stretch the vane to one side or the other out of plane, much like the auto-cambering of the pterosaur wing (*Palmer and Dyke, 2012*). The associated, rapid, stretching and release this would cause is known as aeroelastic flutter (*Lillico et al., 1997*), and it produces extremely high drag coefficients. Drag on the tail vane would therefore be substantially reduced if the tail vane had a mechanism of tensioning.

Our investigation of the tail vane under LSF revealed a lattice of two sets of structures: one set are thicker, regularly spaced tube-like structures with a vertical long axis (*Figure 2*). The second set are thinner, more numerous subhorizontal fibres that criss-cross the thick vertical structures (*Figure 2*). We propose that they both worked as tensile-loaded structures similar to actinofibrils in the wings, but acting in tandem due to the lattice cross-links to resist stretching in multiple directions. This is analogous to how a fabric with fibres running in multiple directions resists stretching. The tail vane would have stretched in the direction of mediolateral deviation. Since the surface would be stretched, the distances between the lattice fibres would increase - pulling, especially, the thicker vertical structures apart. However, these movements would have been limited by both the bending strength of these relatively large vertical 'tubes' and, more importantly, the cross-linking subhorizontal fibres. Since the subhorizontal fibres linked across the vertical structures, any stretching of the vane that spread the vertical structures apart from each other would stretch the subhorizontal fibres along their long axis – loading them in tension. This load would ultimately be taken up at the tail tip, where the fibres converged. This loading regime is consistent with the concavities along the vane edge being aligned with each structure tip, indicating that, under load, the spaces between the structures stretched until the outer edge was linear (*Figure 2D*), with the fossilised position preserving the unloaded slack condition (*Figure 2D*). We see no evidence of active tensioning capacity using muscle and therefore posit that this soft tissue lattice was the primary stiffening adaptation for the tail vane. We are unaware of any extant organisms with an analogous functional feature.

As the precise material composition of the preserved lattice remains elusive, we do not know the absolute value of E for the observed structures, and therefore, cannot calculate expected absolute deflections of the tail vane. However, given the orientation and apparent density of the structures, we can confidently predict that the presence of the lattice would increase the overall elastic modulus of the tail vane in tension, probably substantially. For perspective, even if all of the observed soft tissue was collagen in life, thicker, more condensed collagen fibres can have a tensile E 3.6 times that of thinner, less compact fibres (*Wenger et al., 2007*). To be visibly distinct under LSF, the density and thickness of the structures in the lattice must have been several times that of the surrounding soft tissue matrix, so it is reasonable to surmise that the lattice could have stiffened the vane by a large margin - probably an order of magnitude, at least.

Thus, our results suggest that the tail vane maintained effective stiffness dynamically with internal tension of a cross-linked lattice that minimised excessive vane flutter and associated drag production.

This structural integrity would have permitted the vane to be an effective display structure without incurring excessive aerodynamic costs (*Figure 2E*). This tensioning would also have allowed the tail, as a whole, to be used for mass-shifting-based aerodynamic control without incurring adverse effects of a fluttering vane during rapid tail motions. Tail vane shape changes through ontogeny (*Bennett, 1995*; *O'Brien et al., 2018*) and between species (*Marsh, 1882*; *Döderlein, 1929*; *Frey et al., 2003*), which underscores the importance of the tail vane in early pterosaur evolution. As pterodactyloid pterosaurs evolved a shorter body plan with an anterior center of mass and large heads with cranial crests as the primary display structures (*Jagielska and Brusatte, 2021*), both the control and display functions of the tail were absorbed by the wings and head.

The new soft tissue information also provides clues about the evolutionary origins of the tail vane itself. The cross-linked lattice recognised in this study suggests that the tail vane of early pterosaurs developed from a single contiguous structure rather than a combined structure of scales or feather-like integuments. While the undulating vane edge (*Figure 2D*) might reflect an epidermal patterning, the internal part of the tail vane was likely filled with connective tissue underneath the epidermal layer. The medial part of the tail vane, namely the periphery of the caudal vertebrae, has a different tone under fluorescence and the vertical structures lose clarity when compared to the lateral part of the tail vane (*Figure 2A and B*). This potentially indicates a thicker subdermal connective tissue surrounding the caudal vertebrae. Therefore, the tail vane of pterosaurs consisted of bilateral fleshy folds on the end of the tail, comparable to the cetacean fluke that envelopes dense connective tissue (*Gavazzi et al., 2024*). The growth series of the tail vane shape in *Rhamphorhynchus muensteri* begins with an extended teardrop/oval shape, becoming diamond-shaped (*Figure 2A and B*), and eventually triangular (*Bennett, 1995*). These three shapes parallel the shape changes of the cetacean fluke during embryonic development (*Stěrba et al., 2000*). It is possible that both the pterosaur tail vane and the cetacean fluke evolved through a shared developmental mechanism, perhaps a co-option of the signaling pathway that drives appendage outgrowth (*Gavazzi et al., 2024*), eventually bringing about improved fluid dynamics of the limbs.

## Materials and methods

Over 100 Solnhofen pterosaur fossils were examined for well-preserved tail vanes using an ultraviolet torch at the Bayerische Staatssamlung für Paläontologie (Munich), Museum für Naturkunde (Berlin), Jura Museum (Eichstätt), Natural History Museum (London), National Museum of Scotland (Edinburgh), and Royal Ontario Museum (Toronto) representing the pterosaur inventories in their care. These specimens were screened without bias regardless of their ontogenetic stage and preservation (i.e. completeness, degree of articulation and the presence or absence of soft tissues under white light). Out of this significant sample, only four specimens were found to preserve tail vane soft tissues, which were then imaged under Laser-Stimulated Fluorescence (LSF; *Kaye et al., 2015*) at the Natural History Museum, National Museum of Scotland, and the Royal Ontario Museum. LSF involved projecting a 405 nm violet laser diode from a line lens and scanning it over the specimens in a dark room following standard laser safety protocol. Long exposure photographs over 30 s were taken with a Nikon digital single-lens reflex camera fitted with a 425 nm laser blocking filter. LSF images were then postprocessed for equalisation, saturation, and colour balance across the entire images in Adobe Photoshop CS6.

## *Rhamphorhynchus muensteri* NHMUK PV OR 37003

Initially described by *von Meyer, 1846* as *R. gemmingi*; followed by *R. meyeri* (*Owen, 1870*), described under Nr. 39 in *Wellnhofer, 1975*; synonymised under the same species by *Bennett, 1995*. No keratin appears conserved on pedal digits or rhamphotheca, despite good preservation. Wellnhofer in his assessment does not mention the presence of a tail vane and the feature is not discernible in visible light. Detected with UV and then imaged more vividly under LSF. The specimen is better articulated than NHMUK PV OR 37787. Ontogenetically, on basis of the size of preserved elements and tail vane morphology, it likely belongs to a similar growth stage to NHMUK PV OR 37787. The distal end of tibia is unfused, pointing at immaturity of the animal. The slab contains a ventrally exposed lower jaw and heavily disarticulated postcranial material. It preserves fragments consisting of disarticulated flight apparatus and partially preserved scapula-coracoid. 11 rib-bearing dorsal vertebrae in articulation; disarticulated unfused pubic elements, both hindlimb elements with paired well-preserved tarsals in

articulation. Metacarpal IV is 23 mm long. The wing phalanges make up an incomplete ~380 mm long wing (WP1 = ~113 mm but incomplete, WP2 = ~112 mm but incomplete, WP3 = ~101 mm, WP4 = ~54 mm but incomplete). The tibia is 52 mm long and the incomplete femur ~32 mm long. Metatarsals 1–5 are 26 mm, 28 mm, 27 mm, 21 mm, and 10 mm long, respectively. The preserved tail vane is slightly shorter than lower jaw (89 mm but incomplete), with two features being similar sized. The total tail length is 320 mm, it is preserved with filiform and almost entirely intact, only disarticulating anteriorly. The terminal tail vane is 70 mm of the tail length (22%). The vane encompasses over fifteen vertebrae and is 41 mm wide.

### *Rhamphorhynchus muensteri* NHMUK PV OR 37787

Initially assigned to *Rhamphorhynchus muensteri* by *Goldfuß, 1831*; followed *by R. gemmingi* by *von Meyer, 1846*; then re-assessed under label Nr. 42 in *Wellnhofer, 1975*; cumulatively reassigned to *R. muensteri* by *Bennett, 1995* who synonymised all *Rhamphorhynchus* species as different year classes of the same species. Wellnhofer already noted the faint preservation of a tail vane in his assessment. The slab contains largely disarticulated postcranial material. The skull is partially preserved but severely deformed and eroded, with an outline of the paired ramus (70 mm rami length) and upper crania, retaining anterior dentition with larger teeth (14 mm) displaced anteriorly to the preserved jaws. Most postcranial material is composed of the disarticulated ribcage, with scattered ribs and a partially articulated dorsal vertebral segment. Element of the very deformed sternum and paired scapula-coracoids are visible along with a humerus. The humerus is 39 mm long, a stocky bone with deflecting deltopectoral crest characteristic of *R. muensteri*. The radius is 71 mm long and fourth metacarpal is 24 mm long. The wing phalanges make up a~465 mm long wing (WP1 = 123 mm, WP2 = ~119 mm, WP3 = ~112 mm, WP4 = 111 mm), accumulating to a wingspan of ~1.2 m, with the terminal phalange retaining a 165° curvature. The vertebral column is preserved as modular chunks in partial articulation, composed of presacral vertebrae, sacral vertebrae and over 30 caudal vertebrae in partial articulation; the filiform of the caudal section disarticulate and splay anteriorly. Orientation of preservation enables good visibility of individual caudal bones, as in some specimens these are obscured by elongate, onlapping chevrons and zygapophyses. The total tail length is 362 mm. The vane makes up 75 mm of the tail (21%) and is 41 mm at its widest point.

### *Rhamphorhynchus muensteri* NMS G.1994.13.1

The specimen was assessed by Wellnhofer (pers. comm.) and mentioned in *Frey et al., 2003* for the likelihood of preserving a 'throat pouch'. The specimen was later used in 'tail fin' calculations by *O'Brien et al., 2018*, but remained formally undescribed. While retaining a well-preserved tail vane, the wing membrane, despite discolouration and topographic lineation, does not preserve actinofibrils. Keratinous terminations of rhamphotheca and unguals are also absent. The skull has a proportionally sizeable orbit while the wing phalange extensor process and pubic fusing point to a late-stage sub-adult ontogenetic stage. The slab contains a largely complete and modularly articulated cranial and postcranial skeleton. The skeleton is preserved contorted, seemingly 'biting its tail' with cervical and dorsal vertebrae falling severely out of articulation. Its well-preserved skull, including the sclerotic ring, is 112 mm long with a 86 mm long lower jaw. The scapula-coracoid pair are preserved separately, with forelimb elements (humerus, radius, ulna, metacarpal, manual phalanges) severely deformed or missing. The sterna is ~30 mm. Wings preserve all phalanges, measuring ~467 mm in total length (WP1 = ~130 mm; WP2 = ~134 mm; WP3 = ~125 mm; WP4 = ~78 mm). Its femur and metatarsals are obscured, with only 64 mm measurable on the tibia. While experiencing some deformation, the tail stems from the terminal sacrum and is fully preserved in articulation. The tail measures 348 mm, out of which ~72 mm is taken up by the tail vane. Making up 21% of the total tail length. The tail vane is 'stockier' in comparison to the NHMUK specimens, having a wider anterior opening and measuring 55 mm at its widest section. The vane encompasses at least 15 caudal vertebrae.

## Acknowledgements

Mike Day, Nick Fraser, David Evans and Kevin Seymour are thanked for granting study access to specimens in their care. Michael Pittman was supported by The Chinese University of Hong Kong. Natalia Jagielska was supported by NERC E4DTP studentship NE/S007407/1.

# Additional information

## Funding

| Funder | Grant reference number | Author |
|--------|------------------------|--------|
| Chinese University of Hong Kong | | Michael Pittman |
| Natural Environment Research Council | NE/S007407/1 | Natalia Jagielska |

The funders had no role in study design, data collection and interpretation, or the decision to submit the work for publication.

## Author contributions

Natalia Jagielska, Conceptualization, Data curation, Formal analysis, Validation, Investigation, Visualization, Methodology, Writing – original draft, Writing – review and editing; Thomas G Kaye, Conceptualization, Resources, Data curation, Software, Formal analysis, Validation, Investigation, Visualization, Methodology, Writing – original draft, Writing – review and editing; Michael B Habib, Formal analysis, Validation, Investigation, Methodology, Writing – original draft, Writing – review and editing; Tatsuya Hirasawa, Formal analysis, Investigation, Writing – review and editing; Michael Pittman, Conceptualization, Resources, Data curation, Software, Formal analysis, Supervision, Funding acquisition, Validation, Investigation, Visualization, Methodology, Writing – original draft, Project administration, Writing – review and editing

## Author ORCIDs

Natalia Jagielska ⬤ https://orcid.org/0000-0001-7602-5878
Thomas G Kaye ⬤ https://orcid.org/0000-0001-7996-618X
Tatsuya Hirasawa ⬤ https://orcid.org/0000-0001-6868-3379
Michael Pittman ⬤ https://orcid.org/0000-0002-6149-3078

Reviewer #1 (Public review): https://doi.org/10.7554/eLife.100673.3.sa1
Author response https://doi.org/10.7554/eLife.100673.3.sa2

---

# Additional files

## Supplementary files
• MDAR checklist

## Data availability

All data generated or analysed during this study are included in the manuscript, specifically in the main text and in the high-resolution photos and drawings provided in Figure 2.

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
