## [Editor Report · eLife Assessment]

The presented soft tissue data of pterosaur tail vanes represent a **valuable** contribution to ongoing research efforts to decipher the flight abilities of pterosaurs in the fields of paleontology, comparative biomechanics, and bioinspired design. The new methods are **compelling** and give new detail on tail morphology, with a potential to resolve how pterosaurs were able to control and maintain tail stiffness to furnish flight control.

---

## [Referee Report · Reviewer #1 (Public review)]

This paper reports fossil soft-tissue structures (tail vanes) of pterosaurs, and attempts to relate this to flight performance and other proposed functions for the tail

The paper presents new evidence for soft-tissue strengthening of vanes using exciting new methods.

There is now some discussion of bias in the sample selection method as well as some theory to show how the lattice could have functioned, other than a narrative description.

---

## [Author Response]

The following is the authors’ response to the original reviews.

**Public Reviews:**

**Reviewer #1 (Public review):**
Summary:This paper reports fossil soft-tissue structures (tail vanes) of pterosaurs, and attempts to relate this to flight performance and other proposed functions for the tailStrengths:The paper presents new evidence for soft-tissue strengthening of vanes using exciting new methods.

We thank Reviewer #1 for the positive assessment of our work.

Weaknesses:There seems to be no discussion of bias in the sample selection method - even a simple consideration of whether discarded specimens were likely not to have had the cross-linking lattice, or if it was not visible.There seems to be no supporting evidence or theory to show how the lattice could have functioned, other than a narrative description. Moreover, there is no comparison to extant organisms where a comparison of function might be drawn.

We note these weaknesses and have addressed them as part of the consensus of suggested edits given below (‘first option’). We thank the reviewer for this feedback.

**Reviewer #2 (Public review):**
Summary:The authors have set out to investigate and explain how early members of the Pterosauria were able to maintain stiffness in the vane of their tails. This stiffness, it is said, was crucial for flight in early members of this clade. Through the use Laser-Stimulated Fluorescence imaging, the authors have revealed that certain pterosaurs had a sophisticated dynamic tensioning system that has previously been unappreciated.Strengths:The choice of method of investigation for the key question is sound enough, and the execution of the same is excellent. Overall the paper is well written and well presented, and provides a very succinct, accessible and clear conclusion.

We thank Reviewer #2 for their positive assessment of our work.

Weaknesses:None

We thank Reviewer #2 for their positive assessment of our work.

**Recommendations for the authors:**
The consensus between the reviewers and reviewing board is that this manuscript can be substantially strengthened and this can be achieved in two ways that are presented in order of preference.First option; resolve the following weaknesses:- Include a rigorous discussion of possible bias in the sample selection method with consideration of discarded specimens in relation to cross-linking lattice observation.- Include published biomechanics theory, supported by citations or a self-derived biomechanical model, to show how the lattice could have functioned biomechanically.- Discuss whether you found similar mechanisms in extant organisms for comparative functional interpretation.

We thank the reviewers and reviewing board for taking the time to discuss the review and propose two consensus options for how to substantially strengthen the manuscript. We carefully considered both proposed options and decided to implement the first option in full. We have therefore made main text edits relating to all three points of the first option. The marked up article file shows exactly which parts of the text were edited in relation to the points.

Second option; rewrite the manuscript so no mechanistic claims are made that are not supported by the information presented:- Accept the possibility of sampling bias and its limitation in the presentation of cross-linking lattice observation, outlining future work needed to address this.- Discuss biomechanics theory needs to be developed to show how the lattice could have functioned biomechanically and remove unsupported speculation about this. It is acceptable to present a new hypothesis, clearly outline the motivation for the hypothesis and how it can be tested with future biomechanical and comparative studies. Remove and replace all current speculative sections and phrasing accordingly and replace this with the framework supporting the idea of a new hypothesis.

The first option was implemented instead of the second option.